# Free and Bound Volatile Aroma Compounds of ´Maraština´ Grapes as Influenced by Dehydration Techniques

**Irena Budić-Leto [1,*], Iva Humar [1], Jasenka Gajdoš Kljusurić [2], Goran Zdunić [1] and Emil Zlatić [3]**

[1] Institute for Adriatic Crops and Karst Reclamation, Put Duilova 11, 21 000 Split, Croatia; Iva.Humar@krs.hr (I.H.); goran.zdunic@krs.hr (G.Z.)

[2] Faculty of Food Technology and Biotechnology, University of Zagreb, Pierottijeva 6, 10 000 Zagreb, Croatia; jgajdos@pbf.hr

[3] Department of Food Science and Technology, Biotechnical Faculty, University of LjubljanaJamnikarjeva 101, 1000 Ljubljana, Slovenia; emil.zlatic@bf.uni-lj.si

* Correspondence: irena.budic-leto@krs.hr; Tel.: +385-21-434-420

**Abstract:** Dehydration or drying of grapes is one of the most important steps in the production of Croatian traditional dessert wine Prošek. The natural sun drying of grapes is the traditionally used method in Prošek production. Alternative methods, such as dehydration under controlled conditions, have been studied as safer and faster methods than the traditional sun drying but without precise knowledge of the effect on volatile compounds. The objective of this work was to study how dehydration of grapes carried out in a greenhouse and an environmentally controlled chamber impacts on the free and glycosidically bound volatile compounds of native grape cv. 'Maraština'. The 36 volatile compounds were identified and quantified using headspace solid-phase micro extraction coupled with gas chromatography-mass spectrophotometry (HS-SPME-GC/MS). The results showed that the aroma profile of dehydrated grapes was significantly different from that of fresh grapes. Regarding free forms, significant increases in the concentration of 2-methyl-1-propanol, 1-butanol, 2-hexen-1-ol, 1-hexanol, ethyl hexanoate, hexyl acetate, o-cymene, linalool oxide, and terpinen-4-ol and geraniol were found in greenhouse-dried grapes, whereas increases in cis-limonene-epoxide, trans-limonene epoxide, and γ-hexalactone were higher in chamber-dried grapes compared to greenhouse-dried grapes. Glycosidically bound forms of o-cymene, linalool oxide, linalool, and terpinen-4-ol were increased in both types of drying, whereas β-damascenone was increased only in greenhouse-dried grapes.

**Keywords:** free and bound volatiles; varietal aroma; terpenes; β-damascenone; drying

## 1. Introduction

In the coastal Croatian region Dalmatia, off-vine dried grapes characterized by their high sugar content are used for making the traditional naturally sweet wine Prošek. The most common technique to increase the sugar content in grapes is their exposure to sun for partial dehydration, such as in the production of naturally sweet Passito wines [1] or in Mediterranean dessert wines, such as 'Pedro Ximénez' [2]. Dehydration of the grapes by sun drying is a natural process strongly dependent on the environmental conditions; several factors can affect the quality of the grapes, such as insect attack, intense solar radiation, and the growth of fungi that can produce toxins, such as ochratoxin A [3]. Chamber-drying techniques in controlled conditions as an alternative to traditional sun drying revealed a significant impact of the dehydration rate and temperature on the modification of volatile compounds from the grapes [4–6]. Grape dehydration causes substantial water loss and a gradual increase in sugar concentration, which promotes the synthesis of metabolites that lead to the formation of the typical

flavor of the sweet wines [4,7–10]. Therefore, grape dehydration is one of the most important steps in the production of naturally sweet wine Prošek that influences the production of key volatile compounds in the grape variety used.

Several chemical classes of compounds arising from grapevine metabolism, such as terpenes, norisoprenoids, benzenoids, and C6 alcohols, are responsible for the aroma and flavors of the grapes. These compounds can be found in either free volatile or odorless glycosidically bound form [11,12]. Glysosidically bound compounds have been identified in many grape varieties as grape precursors responsible for some of the specific varietal aroma attributes of the wines because they can be hydrolyzed to volatile aroma compounds by the action of glycosidase enzymes or due to acidic conditions in grape juice/wine [8,11,12]. Despite the components belonging to the varietal aromas being typically found in low concentrations in wines compared to fermentation aromas, they are powerful odorants due to their very low sensory perception, especially free monoterpenes, which are responsible for the floral and fruity aromas [12,13].

*Vitis vinifera* grape varieties belonging to the Muscat group and other terpenol-related varieties, such as 'Riesling' or 'Gewürztraminer', are characterized by the highest concentration of terpenols at the level above the sensory threshold, namely linalool geraniol, nerol, and $\alpha$-terpineol, whereas non-Muscat or "neutral" varieties are mainly characterized by the presence of alcohols and aldehydes although terpenes are present in their juice [12,14,15]. The biosynthesis of aroma compounds in grapes is a highly complex process that includes many pathways and chemical reactions [16,17]. In addition, the grape metabolism is changed after harvest from aerobic to anaerobic, which impacts on the aroma compounds from a grape variety [9,18].

Croatian white native grapes of 'Maraština' (*Vitis vinifera* L.) have been used for making the high-quality, sweet, and traditional Prošek wine in Dalmatia for a long time. 'Maraština' is considered to have an excellent oenological potential for drying, but no data on the grape volatile composition of this variety exist in the literature. The aim of this research was to identify and characterize free and bound volatile aroma compounds in the 'Maraština' grapes and to determine the influence of grape dehydration carried out in the greenhouse and chamber conditions on the volatiles. This is the first detailed research study on the free and glycosidically bound volatile compounds in the grape of cv. 'Maraština'. This work contributes to the understanding of the varietal grape aroma and to improving the quality of naturally sweet wine Prošek.

## 2. Materials and Methods

### 2.1. Greenhouse and Chamber Drying Experiments

The original technology for Prošek wine making involves drying the grapes; however, different drying methods are currently used in Croatia (sun-drying, chamber-drying, greenhouse-drying, etc.). To compare volatile compounds formed in dried grapes under two controlled conditions, we set up a drying experiment in the greenhouse that is a different technique from the direct sun-drying method and the chamber-drying one.

The experiments were conducted with the native grapes of 'Maraština' cultivar (*Vitis vinifera* L.) grown on the peninsula Pelješac, Croatia (Dalmatia). A total of 300 kg of grapes were manually harvested at the technological maturity and partially dried in a single layer under the controlled greenhouse conditions (Schwarzmann, volume of 225 m$^3$). Only grapes with visually good health status were used for the experiment. Daily temperature in the greenhouse was between 17 and 37 °C, and indoor humidity was 55 ± 13%.

At the same time, a total of 20 kg of grapes were uniformly distributed in a single layer in a hot-air chamber SP-440 C (Kambič, Semič, Slovenia, internal dimension 1000 (W) × 800 (H) × 550 (D) in mm) at a constant temperature of 50 °C ± 0.4 °C in two repetitions. Based on the study of Serratosa et al. [19], chamber-drying at 50 °C had advantages over the sun-drying and chamber-drying at 40 °C in terms of the aroma, flavor, and color of the must from 'Pedro Ximénez' grapes and therefore it was selected as

the investigated temperature of the chamber-drying experiment in the study. The grapes were dried until the sugar content reached around 36 °Brix, which took 22 days in the greenhouse, and 3 days in the chamber. Randomly chosen samples of grape were collected from at least 10 different points every other day in the greenhouse and every day in the chamber to determine the progression of grape drying. Three representative samples of grapes were collected for the analyses of aroma compounds at the beginning and at the end of the dehydration process and squeezed by horizontal presses to obtain juice. Analysis of volatile compounds was performed one month after storage of juice samples at −80 °C.

## 2.2. Determination of Basic Grape Parameters

The sugar content in °Brix was measured using a refractometer (Master Baume 2594, Atago, Japan). The pH was measured using a pH meter Titrino 718, (Metrohm, Switzerland). Basic chemical parameters were determined according to the reference OIV methods for wine analysis [20]. Total phenols were determined according to the method of Singleton and Rossi [21].

## 2.3. Analysis of Free and Bound Volatile Aroma Compounds

Headspace Solid-Phase Microextraction—Gas Chromatography/Mass Spectrometry (HS SPME—GC/MS Analysis)

Free fraction. A volume of 10 mL of juice, 10 µL of 4-methyl-2-pentanol (internal standard, 1 mg/L standard solution in methanol, Merck, Darmstadt, Germany), and 3 g NaCl (p.a.) were added into a 20-mL vial. The vial was sealed with an aluminum cover and Teflon-lined cap, and the sample was pre-conditioned in a temperature-controlled heating module at 45 °C for 30 min and agitated at 350 rpm. Volatile aroma compounds were isolated using headspace solid-phase microextraction (HS-SPME) according to the modified method of Castro et al. [22] and analyzed by gas chromatography/mass spectrometry. SPME fibers coated with CarboxenTM/polydimethylsiloxane (1 cm long, 85 µm thick) were purchased from Supelco (Bellefonte, PA, USA). For desorption, the fiber was inserted into a GC/MS injector port at 270 °C in the splitless mode for 2 min.

Identification and quantification of volatile compounds was performed using a gas chromatograph (Agilent 7890 Series GC system) coupled to a 5975C mass selective detector (Agilent, Santa Clara, CA, USA) equipped with a MPS2 Multipurpose autosampler (Gerstel, Baltimore, USA). A DB-5 capillary column was used (60 m × 0.32 mm i.d., with 1 µm film thickness) (Restek, Bellefonte, PA, USA). The initial oven temperature was held at 40 °C for 2 min, then raised at 5 °C /min up to 150 °C, held at 150 °C for 5 min, raised to 250 °C at 5 °C/min, and held at 250 °C for 5 min. Helium was the carrier gas with a flow rate of 1.3 mL/min at 40 °C. The detector operated in the m/z range between 30 and 250. The ion source and quadrupole temperature were maintained at 250 and 150 °C, respectively. Identification of compounds was performed by comparing retention times and mass spectra with those of available commercial standards and with mass spectra from the NIST mass spectral database (National Institute of Standards and Technology, USA). When standards were not available, compounds were tentatively identified by comparing their mass spectra with those in the NIST library. Tentative identification was considered successful for compounds with the MS spectra match probability higher than 80.0%. Standard calibration curves were constructed for quantification of compounds. Results were expressed as the concentration of compounds calculated from the peak area of the individual compounds in comparison to the peak area of internal standard. The repeatability of the experimental method was determined by performing three replicate analyses of each juice sample. Calculated relative standard deviations (RSD %) of peak areas were less than 10%.

Bound fraction. All compounds in the juice were extracted by mixing with 2 M citric acid and subjected to acidic hydrolysis (pH 2.5) according to the modified method described by Pendroza et al. [23]. After adding internal standard, 10 mL of hydrolyzed extract were stirred by the CarboxenTM/polydimethylsiloxane (1 cm long, 85 µm thick)-coated fiber at 45 °C for 30 min.

Both free and bound compounds extracted onto the fiber in this procedure, and the bound fraction of each compound was obtained by subtracting its corresponding free concentration.

### 2.4. Data Analysis

Based on the studies of Budić-Leto et al. [24] and De Villiers et al. [25], univariate characterization was conducted using one-way ANOVA to establish which compounds differed significantly between three groups: grape before drying (fresh grapes) and two drying treatments. Multivariate analyses were also performed: (i) factor analysis (FA) and (ii) principal component analysis (PCA). In FA, the selection of parameters was set to the factor pattern of 0.7 for the first two factors. This decreased the initial set of 46 parameters (40 volatiles and total concentration of 6 chemical classes) to 23. This reduced set was used for the PCA analysis to obtain maximum information from the extracted PCs. STATISTICA, version 8.1 (Statsoft Inc., Tulsa, OK, USA) was used to conduct all data analyses.

## 3. Results and Discussion

### 3.1. Major Quality Parameters of 'Maraština' Grapes

The basic quality parameters of 'Maraština' juices from fresh and both types of partly dried grapes are shown in Table 1.

**Table 1.** Basic parameters of juices from 'Maraština' fresh grapes, greenhouse-dried (G-D) grapes, and chamber-dried (C-D) grapes at 50 °C (average ± standard deviation).

|  | Fresh Grapes | G-D Grapes | C-D Grapes |
|---|---|---|---|
| °**Brix** | 21.2 ± 1.2 [a] | 36.2 ± 0.1 [b] | 35.6 ± 5.9 [b] |
| **pH** | 3.6 ± 0.2 [a] | 4.0 ± 0.0 [b,#] | 4.3 ± 0.1 [c,#] |
| **total acidity** (g/L) | 5.5 ± 0.4 [a] | 4.2 ± 0.1 [b,#] | 6.4 ± 0.6 [#,c] |
| **total phenols** (mg/L) | 335 ± 157 [a] | 631 ± 22 [b] | 783 ± 137 [b] |

Different letters within the row (a, b, c)—indicate significant differences, $p < 0.05$. [#] statistically significant differences in juice must properties between the grape drying methods, $p < 0.05$.

The drying process was stopped at the same sugar concentration in the greenhouse- and in chamber-dried grapes. Sugar content increased to 36.2 °Brix in greenhouse-dried grapes after 22 days (524 hours), and in chamber-dried grapes to 35.6 °Brix in 3 days (72 hours). The concentration process resulted in significant increases in sugar content, total phenols, and pH value. These results are in agreement with previously published reports [5,26,27].

### 3.2. Free Volatile Aroma Compounds in Fresh and Dehydrated Grapes

Free volatile compounds in the fresh grapes and in the grapes after dehydration are shown in Table 2 grouped in the chemical classes.

A total of 36 volatile compounds were identified in the fresh grapes, including alcohols (8), esters (8), terpenes (13), ketones (2), aldehydes (3), C-13 norisoprenoids (1), and lactone (1). Regarding free volatile compounds, 1-hexanol, 2-hexen-1-ol, and 3-methyl-1-butanol occurred in the highest concentrations in fresh grapes, followed by ethyl acetate and 2-hexanal. Linalool and linalool oxide were present in the highest concentration among terpenols, but their concentration was quite low, 3.81 and 3.04 µg/L, respectively. Other identified terpenes, such as α-terpineol, myrcene, o-cymene, terpinen-4-ol, and geraniol, were found in concentrations between 0.25 and 0.38 µg/L, whereas citronellol, nerol, linalool acetate, and geranyl acetone, were found in concentrations between 0.03 and 0.08 µg/L. Both aromatic alcohols (2-phenylethanol and benzyl alcohol) were also detected in the free fraction, although in very low concentrations.

**Table 2.** Average values (±standard deviation) of free volatile compounds in fresh grapes, greenhouse-dried (G-D) grapes, and chamber-dried (C-D) grapes at 50 °C. The significant differences between the drying method and grapes before drying are given as *p*-values.

| Compound | Abbreviation | Concentration (µg/L) | | | *p*-Value |
|---|---|---|---|---|---|
| | | **Fresh Grapes** | **G-D Grapes** | **C-D Grapes** | |
| 2-methyl-1-propanol | A1 | 0.26 ± 0.12 [a] | 0.78 ± 0.06 [b] | n.d. | <0.05 |
| 1-butanol | A2 | 0.53 ± 0.17 [a] | 3.45 ± 0.07 [b] | 0.57 ± 0.34 [a] | <0.05 |
| 3-methyl-1-butanol | A3 | 17.11 ± 7.51 [a] | 42.54 ± 1.57 [b] | 28.06 ± 20.83 [a] | 0.281 |
| 2-methyl-1-butanol | A4 | 0.72 ± 0.27 [a] | 1.59 ± 0.05 [b] | 0.89 ± 0.67 [a] | 0.144 |
| 2-hexen-1-ol | A5 | 21.98 ± 10.16 [a] | 56.74 ± 2.08 [b] | 0.24 ± 0.21 [b] | <0.05 |
| 1-hexanol | A6 | 50.68 ± 8.97 [a] | 89.16 ± 6.76 [b] | 0.46 ± 0.65 [b] | <0.05 |
| benzyl alcohol | A7 | 0.18 ± 0.06 [a] | 0.11 ± 0.01 [a] | 0.59 ± 0.04 [b] | <0.05 |
| 2-phenylethanol | A8 | 0.03 ± 0.02 [a] | n.d. | 0.11 ± 0.15 [a] | 0.272 |
| Σ alcohols | Alcohols | 91.49 ± 10.57 [a] | 194.37 ± 9.78 [b] | 30.93 ± 21.58 [b] | <0.05 |
| ethyl acetate | E1 | 14.21 ± 5.84 [a] | 21.68 ± 0.35 [a] | 26.98 ± 6.6 [a] | 0.226 |
| ethyl lactate | E2 | 0.46 ± 0.07 [a] | 0.35 ± 0.01 [a] | 0.32 ± 0 [a] | <0.05 |
| isoamyl acetate | E3 | 1.04 ± 1.44 [a] | 0.23 ± 0.09 [a] | 0.19 ± 0.03 [a] | 0.629 |
| ethyl hexanoate | E4 | 0.06 ± 0.03 [a] | 0.25 ± 0.04 [b] | 0.09 ± 0.02 [a] | <0.05 |
| hexyl acetate | E5 | 0.12 ± 0.02 [a] | 0.46 ± 0.03 [b] | 0.05 ± 0.02 [b] | <0.05 |
| ethyl octanoate | E6 | 0.01 ± 0.02 [a] | n.d. | 0.11 ± 0.03 [b] | <0.05 |
| 2-phenyl-ethyl acetate | E7 | 0.01 ± 0.02 [a] | n.d. | 0.02 ± 0.03 [a] | 0.272 |
| ethyl nonanoate | E8 | n.d. | n.d. | 0.02 ± 0.03 [a] | 0.272 |
| ethyl decanoate | E9 | n.d. | n.d. | n.d. | - |
| Σ esters | Esters | 15.91 ± 6.41 [a] | 22.97 ± 0.31 [a] | 27.78 ± 6.45 [a] | 0.252 |
| α-pinene | T1 | n.d. | n.d. | n.d. | - |
| myrcene | T2 | 0.30 ± 0.11 [a] | 0.15 ± 0.01[a] | 0.15 ± 0.04 [a] | 0.737 |
| o-cymene | T3 | 0.33 ± 0.10 [a] | 1.16 ± 0.03 [b] | 0.15 ± 0.03 [a] | <0.05 |
| linalool oxide | T4 | 3.04 ± 0.96 [a] | 4.77 ± 0.32 [b] | 4.86 ± 0.05 [a] | 0.737 |
| linalool | T5 | 3.81 ± 2.09 [a] | 1.63 ± 0.20 [a] | 1.26 ± 0.35 [a] | 0.213 |
| *cis*-limonene epoxide | T6 | 0.52 ± 0.28 [a] | 0.23 ± 0.19 [a] | 5.93 ± 1.09 [b] | <0.05 |
| *trans*-limonene epoxide | T7 | 2.12 ± 0.84 [a] | 2.54 ± 0.21 [a] | 6.35 ± 1.17 [b] | <0.05 |
| terpinen-4-ol | T8 | 0.24 ± 0.15 [a] | 7.67 ± 0.2 [b] | 0.13 ± 0.10 [a] | <0.05 |
| α-terpineol | T9 | 0.38 ± 0.38 [a] | 0.35 ± 0.03 [a] | 0.36 ± 0.17 [a] | 0.926 |
| citronellol | T10 | 0.05 ± 0.01 [a] | 0.06 ± 0.02 [a] | 0.05 ± 0.06 [a] | 0.636 |
| nerol | T11 | 0.03 ± 0.04 [a] | 0.06 ± 0.01 [a] | 0.04 ± 0.06 [a] | 0.675 |
| linalool acetate | T12 | 0.03 ± 0.02 [a] | n.d. | n.d. | - |

**Table 2.** *Cont.*

| Compound | Abbreviation | Concentration (µg/L) | | | *p*-Value |
|---|---|---|---|---|---|
| | | **Fresh Grapes** | **G-D Grapes** | **C-D Grapes** | |
| geraniol | T13 | 0.29 ± 0.17 [a] | 0.4 ± 0.05 [a] | 0.10 ± 0.01 [a] | <0.05 |
| geranyl acetone | T14 | 0.08 ± 0.03 [a] | 0.03 ± 0.02 [b] | 0.04 ± 0.01 [a] | 0.539 |
| nerolidol | T15 | n.d. | n.d. | n.d. | - |
| Σ terpenes | Terpenes | 11.23 ± 4.14 [a] | 19.06 ± 0.98 [a] | 19.4 ± 3.1 [a] | 0.859 |
| 2-hexenal | Ald-1 | 5.19 ± 1.15 [a] | 5.61 ± 0.4 [a] | 0.18 ± 0.03 [b] | <0.05 |
| benzaldehyde | Ald-2 | 1.49 ± 0.48 [a] | 4.31 ± 0.31 [b] | 13.11 ± 8.03 [b] | 0.129 |
| acetal | Ald-3 | 0 ± 0 [a] | 0.07 ± 0 [b] | 0.4 ± 0.46 [a] | 0.265 |
| Σ aldehydes | Aldehydes | 6.68 ± 1.32 [a] | 9.99 ± 0.44 [b] | 13.7 ± 8.52 [a] | 0.471 |
| α-ionone | K1 | n.d. | n.d. | n.d. | - |
| 6-methyl-5-heptene-2-one | K2 | 1.14 ± 0.24 [a] | 0.79 ± 0.01 [a] | 1.24 ± 0.58 [a] | 0.239 |
| 2,3-pentadione | K3 | 0.52 ± 0.1 [a] | 2.29 ± 0.09 [b] | 3.37 ± 3.85 [a] | 0.633 |
| Σ ketones | Ketones | 1.66 ± 0.18 [a] | 3.08 ± 0.1 [b] | 4.61 ± 4.43 [a] | 0.561 |
| β-damascenone | β-dem | 4.01 ± 1.7 [a] | 1.44 ± 0.27 [b] | 0.35 ± 0.19 [b] | <0.05 |
| γ-hexalactone | γ-hex | 0.13 ± 0.03 [a] | 0.08 ± 0.02 [a] | 0.25 ± 0.02 [b] | <0.05 |
| Σ benzenoids (A7+A8+Ald-2) | Benzenoids | 1.69 ± 0.46 [a] | 4.43 ± 0.32 [b] | 13.81 ± 8.22 [b] | 0.119 |

Different letters within the row (a, b)—indicate significant differences, *p* < 0.05 n.d. not detected.

The composition of free volatile compounds found in 'Maraština' grapes was typical for the non-aromatic grapes. The concentrations of free alcohols and terpenes were similar to 'Fiano' grape juice used for the production of dry wine Fiano di Avellino, as well as for the sweet wines [11]. Interestingly, a relatively high concentration of β-damascenone was also detected in the free volatile fraction in 'Maraština' grapes. It is already known that the free form of aroma compounds occur in the non-aromatic grapes in concentrations lower than their sensory threshold. However, several studies have shown a positive correlation between the sensory characteristics of the pool of volatile compounds released from grape glycosylated precursors and the varietal sensory properties of the wines obtained from them [11,12,23].

Dehydration technique significantly impacted the concentration of free volatile compounds found in the dried grapes of 'Maraština'. Significant differences were found for 18 out of 35 volatile compounds. There was a general increasing trend for the concentration of the free volatile compounds of 'Maraština' grapes subjected to dehydration. An increase in concentration of free volatile compounds is expected as a result of water loss by evaporation during partial drying. This is in agreement with the results of Franco et al. on the 'Perdo Ximénez' must of sun-dried grapes.

Dehydration increased the concentration of total alcohols in greenhouse-dried grapes. The ratios among individual alcohols in dried grapes followed a similar trend to those in fresh grapes. Among the volatiles, 1-hexanol, 2-hexen-1-ol, and 3-methyl-1-butanol were the major compounds in greenhouse-dried grapes. The opposite effect was found for chamber-dried grapes, whereby the concentration of C6 alcohols was significantly lower compared to fresh grapes. This differential effect on the concentration of C6 compounds could have been due to different metabolic changes occurring in grapes subjected to a fast dehydration rate at higher temperatures that cause a stronger concentrating effect and a faster water loss.

A recent study by Zenoni et al. [28] found that slow dehydration is necessary to induce gene expression and metabolite accumulation associated with the final quality traits of dehydrated berries. The accumulation of desirable key metabolites during postharvest dehydration is inhibited by rapid dehydration conditions that shorten the berry lifetime. Metabolomics and transcriptomics have demonstrated that the dehydration process results in modulation of the expression of various genes implicated in grape metabolism, such as secondary metabolism of phenols, terpenes, and lipids [10]. The water loss is reflected in the cell metabolism that transitions from aerobic to anaerobic [18,29]. This shift is associated with alcohol dehydrogenase (ADH) activity and may influence the metabolic changes in volatile compounds depending on the temperature, relative humidity, and level of dehydration [10,28]. The activity of oxidative enzymes, such as lipoxygenase (LOX), leads to the formation of C6 alcohols and aldehydes [6,9].

Among esters, ethyl acetate was found in the highest concentration, comprising more than 90% of the total ester concentration in both types of dried grapes. Although significant differences were found for ethyl lactate, ethyl hexanoate, ethyl octanoate, and hexyl acetate between the types of dried grapes, their concentrations were low and varied between 0.05 and 0.32 μg/L.

Terpenes are of great interest to the aroma of wines, and it was important to examine the dehydration effect on their quantitative and qualitative composition in our study. The concentration of monoterpenes significantly increased in dried grapes compare to fresh grapes, although no significant difference in the total terpenes level was found between the two types of dried grapes. Terpinen-4-ol (7.67 μg/L) and linalool oxide (4.77 μg/L) were the prevalent monoterpenes in greenhouse-dried grapes, whereas trans-limonene epoxide (6.38 μg/L) was the major one in chamber-dried grapes. The monoterpene profile is related to the unique sensory aroma of varietal white wine [30]. However, monoterpenes in 'Maraština' dried grapes were found in low concentrations bellow their sensory thresholds, but they can still act as the significant odorants because of their synergy with other compounds [31,32]. The most recent study showed that the composition of non-volatiles strongly increases the volatility of monoterpene isomers in 'Pinot Gris' wine through a combination of enhancing and suppression effects, with the aroma perception changed when more components were present [33].

Monoterpenes were found to influence fruity aromas (orange flowers) in wine, suggesting that linalool (odor threshold of 6 µg/L) [34] and linalool oxide may interact with non-volatiles, resulting in change in their aroma perception [33,35].

The concentration of free β-damascenone in both types of dried grapes significantly decreased in comparison to fresh grapes; their concentration was lowest in chamber-dried grapes. A decrease in the concentration of β-damascenone was also observed with the dehydration of grapes, in agreement with the observations of Slaghenaufi et al. [36].

### 3.3. Bound Volatile Aroma Compounds in Fresh Grapes and Dried Grapes

Concentrations of volatile aroma compounds identified and quantified after acid hydrolysis are presented in the Supplementary Material (Table S1). The concentration of terpene compounds in the fresh grape juice was higher in the bound than free forms. This was quite remarkable considering that terpene compounds are responsible for floral and citrus aromas of wines. In particular, the highest amount was revealed for linalool oxide (5.43 µg/L) and linalool (4.01 µg/L). Linalool represented almost 42% of all terpenes found in bound fraction. Other authors [11] have reported 19% of monoterpene alcohol linalool in the bound form of 'Fiano' grapes. We identified 15 glycosidically bound compounds consisting of three alcohols (1-butanol, 2-hexen-1-ol, and 1-hexanol), six terpenes (myrcene, o-cymene, linalool oxide, linalool, terpinen-4-ol, and linalool acetate), two aldehydes (2-hexenal and benzaldehyde), two ketones (6-methyl-5-heptene-2-one and 2,3-pentadione), and β-damascenone in greenhouse-dried grapes as presented in Table 3.

**Table 3.** Bound fraction of volatile aroma compounds in fresh grapes, greenhouse-dried (G-D) grapes, and chamber-dried (C-D) grapes at 50 °C as average values of three repetitions.

| Compound | Abbreviation | Concentration (µg/L) | | |
|---|---|---|---|---|
| | | Fresh Grapes | G-D Grapes | C-D Grapes |
| 1-butanol | A2 | 0.02 [a] | 0.80 [b] | 0.12 [c] |
| 2-hexen-1-ol | A5 | n.d. | 15.50 [a] | n.d. |
| 1-hexanol | A6 | n.d. | 60.75 [a] | n.d. |
| ΣAlcohols | Alcohols | | | |
| ethyl acetate | E1 | n.d. | 0.91 [a] | 7.87 [b] |
| Σ Esters | Esters | | | |
| α-pinene | T1 | n.d. | n.d. | 0.11 [a] |
| myrcene | T2 | 0.43 [a] | 1.08 [b] | 1.46 [b] |
| o-cymene | T3 | 0.55 [a] | 2.16 [b] | 1.74 [b] |
| linalool oxide | T4 | 5.43 [a] | 14.56 [b] | 49.90 [c] |
| linalool | T5 | 4.01 [a] | 17.11 [b] | 16.99 [b] |
| terpinen-4-ol | T8 | 0.06 [a] | 1.99 [b] | 0.82 [c] |
| linalool acetate | T12 | 0.01 [a] | 0.19 [b] | n.d. |
| Σ Terpenes | Terpenes | | | |
| 2-hexenal | Ald-1 | 0.31 [a] | 2.24 [b] | 0.21 [a] |
| benzaldehyde | Ald-2 | 0.72 [a] | 2.30 [b] | 1.90 [b] |
| Σ Aldehydes | Aldehydes | | | |
| 6-methyl-5-heptene-2-one | K2 | n.d. | 0.16 [a] | 0.23 [a] |
| 2,3-pentadione | K3 | 0.07 [a] | 0.50 [b] | n.d. |
| Σ Ketones | Ketones | | | |
| β-damascenone | β-dem | 15.58 [a] | 67.78 [b] | 11.89 [a] |
| Σ benzenoids (A7 + A8 + Ald-2) | Benzenoids | 0.51 [a] | 2.19 [b] | 1.20 [a] |

Different letters within the row (a, b, c)—indicate significant differences, $p < 0.05$. n.d. not detected

Bound forms of 2-hexen-1-ol, 1-hexanol, and linalool acetate 2,3-pentadione were not found in chamber-dried grapes. Citronellol, nerol, and geraniol were not found in bound forms in fresh nor in dried grapes regardless of the dehydration technique. Early work of Wilson et al. [37] demonstrated that these three monoterpenes never exceeded the glycosidically bound concentration of $\alpha$-terpineol in juice, which was similar to our results because $\alpha$-terpineol was also not found in bound forms in any types of juices.

Compared to terpenes, the concentration of bound $\beta$-damascenone was affected most by the dehydration method, with an increasing concentration in greenhouse-dried grapes. In contrast, in chamber-dried grapes, the concentration of $\beta$-damascenone decreased. $\beta$-damascenone is a norisoprenoid derived from the degradation of carotenoids. An increase in $\beta$-damascenone concentration is expected, as water loss / drying is a stress factor that triggers many physiological changes in grapes (expression of genes that regulate carotenoid biodegradation and aroma compound biosynthesis as reported by Asproudi et al. [38] and Lan et al. [39]). These processes can be triggered even in the freezing process in vine at $-8$ °C, in ice grapes, but the loss of mass (concentration) alone is certainly not a decisive factor in explaining the increase in $\beta$-damascenone in grapes during drying [39]. In our study, we found that the drying temperature (50 °C), despite the concentration effect, negatively impacts and reduces the $\beta$-damascenone concentration in the grapes, perhaps due to the inactivation of key enzymes involved in its precursor biosynthesis. Different precursors of $\beta$-damascenone have been identified in grapes [40] either as glycoconjugates or free norisoprenoids [41]. Acid hydrolysis of glycosidic precursors and subsequent possible rearrangements are likely to be the reason for the increase observed for $\beta$-damascenone.

$\beta$-damascenone is the key aroma compound of sun-dried grapes and the wines made from them [12]. This compound played an important role in the flavor of 'Maraština' grapes, in both types of dried grapes and fresh too. It has quite a low aroma threshold, close to the ng/L, and the aroma of prunes or overripe plums.

In order to determine the influence of the grape drying technique on the differentiation of free and bound volatile components from fresh grapes, we applied principal component analysis (PCA). The PCA showed that 78.96% of the total variance was explained by the first two components (D1 and D2, Figure 1A), whereby the first principal component (D1) explained 46.35% and the second one (D2) 32.61%. The investigated grape samples were separated into three clusters based on the freshness/drying (Figure 1A). The first principal component (D1) showed the domination of loadings for the content of alcohols and aldehydes in greenhouse-dried grapes, regardless of whether it was a free or bound fraction. The second principal component (D2) contained mostly terpenes and esters. In chamber-dried grapes, the dominant bound compounds were linalool, linalool oxide, and ethyl acetate.

The PCA volatile composition distribution (Figure 1B) was informative regarding the changes in the measured parameters. The second quadrant contained the cluster of chamber-dried grapes (Figure 1A), which was related to the highest concentration of 6-methyl-5-heptene-2-one (1.24 µg/L free form) that was dominant in the bound fraction in both types of dried grapes (K2 (b) positioned in the first quadrant). All bound fractions of volatile aroma compounds were scattered in the first, second, and fourth quadrants (Figure 1B) in which dried grapes were positioned (Figure 1A). The presented PCA showed the domination of free fractions of volatile aroma compounds myrcene (T2), linalool (T5), linalool acetate (T12), and $\beta$-damascenone ($\beta$-dem). The clustering of these compounds was in agreement with the results of Slaghenaufi et al. [36].

### 3.4. Evaluation of the Aroma Profile of Dried Grapes of 'Maraština'

To determine the key volatile compounds that impact on the aroma and flavor of the dried 'Maraština' grapes, the odor activity values (OAVs) of major volatile compounds were calculated and presented, together with their odor threshold, aroma descriptors, and aroma series in Table 4.

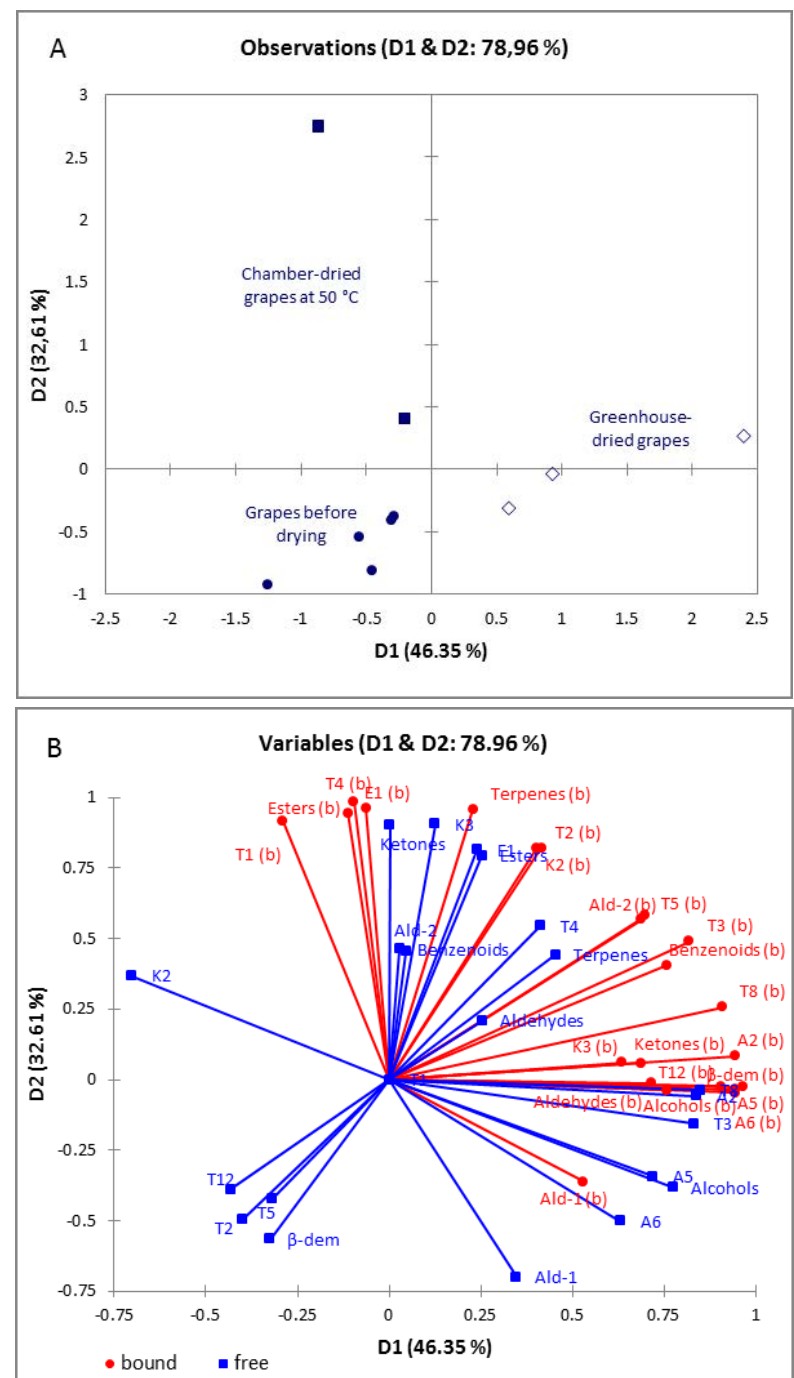

**Figure 1.** Principal component analysis of composition distribution of: (**A**) grapes (fresh or dried) and (**B**) their free and bound (b) volatile compounds.

The contributions of volatiles to the aroma of dried grapes were calculated by dividing the total concentration (free and bound fraction) by the odor threshold of the important volatile compounds. The concentrations of linalool and β-damascenone in greenhouse-dried grapes were far above their odor activity values; therefore, these compounds are very important for the aroma of the dried 'Maraština' grapes. This is in agreement with the Ferreira and Lopez [12].

**Table 4.** Odor threshold (µg/L), aroma descriptor, OAV values assigned to total free and bound key volatile aroma compounds of dried grapes, and aroma series.

| Compound | Odour Threshold (µg/L) | Aroma Descriptor | OAV G-D Grapes | OAV C-D Grapes | Aroma Series |
|---|---|---|---|---|---|
| 1-butanol | 150,000 [a] | Medicinal | 0.00002 | 0.000004 | Solvent |
| 3-methyl-1-butanol | 30,000 [b] | Solvent, sweet cake | 0.001 | 0.0007 | Solvent, sweet |
| 2-hexen-1-ol | 1500 [a] | Green | 0.05 | - | Green, herbaceous |
| 1-hexanol | 1100 [a] | Grass, resinous | 0.14 | - | Fresh, resinous |
| ethyl acetate | 12,000 [a] | Fruity, nail polish | 0.002 | 0.003 | Solvent, fruity |
| ethyl hexanoate | 14 [b] | Apple, banana | 0.025 | - | Fruity |
| hexyl acetate | 670 [a] | Ripe fruit | 0.0001 | - | Floral, fruity |
| linalool oxide | 6000 [d] | Leafy, sweet, floral, creamy | 0.003 | 0.009 | Floral |
| linalool | 6 [e] | Orange flowers | 3.75 | 3.65 | Floral |
| terpinen-4-ol | 5000 [a] | Iris | 0.002 | 0.0002 | Floral |
| 2-hexenal | 9.2 [a] | Green | 0.85 | 0.04 | Fresh |
| benzaldehyde | 2000 [a] | Bitter almond | 0.003 | 0.007 | Nutty, burned |
| β-damascenone | 0.05 [c] | Stewed apple, over-ripe plums | 1384 | 244 | Sweet, fruity |

[a]—As determined in the 1:10 alcohol/water mixture by Franco et al. [2]. [b]—Threshold was determined in synthetic wine consisting of 11% *v/v* ethanol, 7 g/L glycerin, 5 g/L of tartaric acid, pH value 3.4 by Ferreira et al. [13]. [c]—As determined by Guth [42]. [d]—Fariña et al. [43]. [e]—Buttery et al. [34].

## 4. Conclusions

A total of 36 free volatile compounds were identified in the fresh grapes, including alcohols (8), esters (8), terpenes (13), ketones (2), aldehydes (3), C-13 norisoprenoids (1), and lactone (1). Dehydration technique significantly impacted the concentration of 18 free volatile compounds in the juice from dried grapes, among them 2-methyl-1-propanol, 1-butanol, 2-hexen-1-ol, 1-hexanol, ethyl hexanoate, hexyl acetate, o-cymene, linalool oxide, and terpinen-4-ol and geraniol were higher in greenhouse-dried grapes, whereas *cis*-limonene-epoxide and *trans*-limonene epoxide and γ-hexalactone were higher in chamber-dried grapes compared to greenhouse-dried grapes. In grapes subjected to the two drying methods, 15 glycosidically bound volatile compounds were identified. The results of PCA analysis showed that the first two principal components described 78.96 % of the variation among the observed parameters measured in the fresh and dried samples (using the two drying methods), whereby the first principal component (D1) explained 46.35% and the second one (D2) 32.61% of the total variance. The first principal component showed the domination of alcohols and aldehydes in greenhouse-dried grapes, regardless of whether it was a free or bound fraction. The second principal component contained mostly terpenes and esters. In chamber-dried grapes, the dominant bound compounds were linalool, linalool oxide, and ethyl acetate. Results of our study revealed that the key aroma compound β-damascenone, regardless of whether it was a free or bound fraction, had a higher concentration in a slower dehydration rate. These findings will be useful in dessert wine Prošek production. Based on our study, greenhouse-dried grapes at the controlled conditions (max. temperature at 37 °C) may provide an improved must aroma composition for the production of Prošek wine. In opposite to that, our research showed that a chamber-drying temperature of 50 °C negatively influenced the concentration of some alcohols, esters, and especially β-damascenone. Greenhouse-dried grapes will contribute to the aroma profile of sweet and fruity descriptors dominating in the scents of over-ripe plums and stewed apple.

**Supplementary Materials:** The following are available online at http://www.mdpi.com/2076-3417/10/24/8928/s1, Table S1: Average values (±standard deviation) of volatile compounds after hydrolysis (free and bound fractions).

**Author Contributions:** Conceptualization and methodology, I.B.-L. and E.Z.; formal analysis, I.H., E.Z.; investigation, I.B.-L., I.H. and G.Z.; writing—original draft preparation, I.B.-L. and J.G.K.; writing—review and editing, I.B.-L., I.H., J.G.K., G.Z., E.Z.; Supervision I.B.-L. Project Administration I.B.-L. All authors have read and agreed to the published version of the manuscript.

**Funding:** This research received no external funding.

**Acknowledgments:** This study was supported by Ministry of Science and Education of the Republic of Croatia, Project "Biotechnological Parameters of Premium-Quality Dalmatian Dessert Wine-Prošek".

**Conflicts of Interest:** The authors declare no conflict of interest. The funders had no role in the design of the study; in the collection, analyses, or interpretation of data; in the writing of the manuscript, or in the decision to publish the results.

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
