# Peer review of "Free and Bound Volatile Aroma Compounds of ´Maraština´ Grapes as Influenced by Dehydration Techniques"

_applsci, doi:10.3390/app10248928_

Round 1
Reviewer 1 Report
This manuscript entitled "Free and bound volatile aroma compounds of ´Maraština´ grapes as influenced by dehydration techniques" try to explore the role of different dehydratation techniques on the aroma of grapes, and to do so, they have sampled fresh grapes and grapes dried in green-house (is it the same as sun drying? In some countries they are considered different techniques and it must be indicated in the text) as well as grapes dried in a chamber and the free and bound volatile compounds have been analyzed. Author have done a good work and sound analysis of grape aroma, but the manuscript present some shortcomings.
Introduction: LIne 43: Bad use of references, references 4 does not study volatile aroma compounds at all and reference 5 only described sensory analysis but not volatile aroma compounds.
Material and Methods: Line 118: identification of compounds should be expressed as tentative identification.
Results and Discussion
Table 1. There is no statistical letters in the fresh grape column, were these data included in the statistical analysis? Also, the weight of the grape berries should be given in this table.
Table 2: Again, statistical analysis missing for fresh grapes.
Line 233-234: the difference in the concentration of b-damascenone seems to be the most important factor influencing the aroma of chamber-dried or greenhouse-dried grapes but no discussion is given regarding the possible causes for this large difference in concentration.
Line 248: Change "tablica"
Line 290: It is quite surprising that the chamber-dried samples are located very far one from the other, why is the variability so large? Quite similar happens with Greenhouse dried grapes. In fact, some of the repetitions are quite close to grapes before drying
Conclusion: There is no clear conclusion, only a resume of the results. The differences in the concentration of damascenone and some comments of the practical implications of the findings would clearly improve this section.
Author Response
Response to Reviewer 1 Comments
Moderate English changes required
Answer:
The manuscript was revised in English language by native professional speaker.
First reviewer:
This manuscript entitled "Free and bound volatile aroma compounds of ´Maraština´ grapes as influenced by dehydration techniques" try to explore the role of different dehydratation techniques on the aroma of grapes, and to do so, they have sampled fresh grapes and grapes dried in green-house (is it the same as sun drying? In some countries they are considered different techniques and it must be indicated in the text) as well as grapes dried in a chamber and the free and bound volatile compounds have been analyzed. Author have done a good work and sound analysis of grape aroma, but the manuscript present some shortcomings.
Answer:
Question 1: ….is it the same as sun drying? In some countries they are considered different techniques and it must be indicated in the text)
Line 77-81: We added: “The original technology for Prošek wine making involves drying the grapes, however different drying methods are currently used in Croatia (sun-drying, chamber-drying, greenhouse-drying, etc.). To compare volatile compounds formed in dried grapes under two controlled conditions we set up drying experiment in the greenhouse that is different technique from the direct sun-drying method and the chamber-drying one.”
Introduction: LIne 43: Bad use of references, references 4 does not study volatile aroma compounds at all and reference 5 only described sensory analysis but not volatile aroma compounds.
Answer:
Thank you. We agree with the reviewer. We deleted references 4 and 5 and added new references 4 and 5:
Deleted references:
- Serratosa, M.P.; Lopez-Toledano, A.; Medina, M.; Merida, J. Drying of Pedro Ximénez grapes in chamber at controlled temperature and with dipping pretreatments. Changes in the color fraction. J. Agric. Food Chem. 2008, 56, 10739-10746. https://doi.org/10.1021/jf8021767.
- Serratosa, M.P.; Marquez, A.; Lopez-Toledano, A.; Merida, J. Sensory Analysis of sweet musts in Pedro Ximénez cv. grapes dried using different methods. S. Afr. J. Enol. 2012, 33, 14-20. https://doi.org/10.21548/33-1-1301
New references added:
- Ruiz, M.J.; Moyano, L.; Zea, L. Changes in aroma profile of musts from grapes cv. Pedro Ximénez chamber-dried at controlled conditions destined to the production of sweet Sherry wine. LWT-Food Sci Technol. 2014, 59, 560-565. https://doi.org/10.1016/j.lwt.2014.04.056
- Ruiz-Bejarano, M.J.; Castro-Mejías, R.; del Carmen Rodríguez-Dodero, M.C.; García-Barroso, C. Volatile composition of Pedro Ximénez and Muscat sweet Sherry wines from sun and chamber dried grapes: A feasible alternative to the traditional sun-drying. J. Food Sci. Technol. 2016, 53, 2519-2531. https://doi.org/10.1007/s13197-016-2192-1.
Material and Methods: Line 118: identification of compounds should be expressed as tentative identification.
Answer:
Line 123-127: We added: “When standards were not available, compounds were tentatively identified by comparing their mass spectra with those in NIST library”. We added “Tentative identification”… in the front of the sentence …was considered successful for compounds with the MS spectra match probability, higher than 80.0%, and also corrected “identification” to “quantification”.
Results and Discussion
Table 1. There is no statistical letters in the fresh grape column, were these data included in the statistical analysis? Also, the weight of the grape berries should be given in this table.
Answer:
Fresh grape was also included in the statistical analysis, but as we concentrated on the differences in drying, we neglected fresh grape, which is now corrected in Tables 1 and 2.
We are aware that grape berris weight were also modified by dehydration process but we do not think that grape berries should be given in this table because additional details related to materials and methods part (sampling method is quite demanding topic) also need to be add in the text. However, phycal properties of berries (loss of weight, thinness of skin, etc. were not in the focus of the manuscript, only volatiles.
Table 2: Again, statistical analysis missing for fresh grapes.
Fresh grape was also included in the statistical analysis, but as we concentrated on the differences in drying, we neglected fresh grape, which is now corrected in Tables 1 and 2.
Line 233-234: the difference in the concentration of b-damascenone seems to be the most important factor influencing the aroma of chamber-dried or greenhouse-dried grapes but no discussion is given regarding the possible causes for this large difference in concentration.
Answer:
We added:
Line 243-245: “Decreasing in the concentration of β-damascenone was also observed with dehydration of grapes, in agreement with the observations of Slaghenaufi et al. [36].”
Line 272-284: “β-damascenone is a norisoprenoid derived from the degradation of carotenoids. An increase in β-damascenone concentration is expected, as water loss / drying is a stress factor that triggers many physiological changes in grapes (expression of genes that regulate carotenoid biodegradation and aroma compound biosynthesis as reported by Asproudi et al. [38] and Lan et al. [39]. These processes can be triggered even in freezing process in vine at -8 °C, in ice grapes, but the loss of mass (concentration) alone is certainly not a decisive factor in explaining the increase in β-damascenone in grapes during drying [39]. In our study, we found that the drying temperature (50 °C), despite concentration effect, negatively impact and reduces the β-damascenone concentration in the grapes, perhaps due to the inactivation of key enzymes involved in its precursor biosynthesis. Different precursors of β-damascenone have been identified in grapes [40] either as glycoconjugates and free norisoprenoids [41]. Acid hydrolysis of glycosidic precursors and subsequent possible rearrangements are likely to be the reason for the increase observed for β-damascenone.”
We added new references: 38, 39, 40 and 41:
- Asproudi, A.; Petrozziello, M.; Cavalletto, S.; Ferrandino, A.; Mania, E.; Guidoni, S. Bunch Microclimate Affects Carotenoids Evolution in cv. Nebbiolo (V. vinifera L.). Appl. Sci. 2020, 10, 3846.-3864. doi:10.3390/app10113846
- Lan, Y.B.; Qian, X.; Yang, Z.J.; Xiang, X.F.; Yang, W.X.; Liu, T.; Zhu, B.Q.; Pan, Q.H.; Duan, C.Q. Striking changes in volatile profiles at sub-zero temperatures during-over-ripening of ‘Beibinghong’ grapes in Northeastern China. Food Chem. 2016, 212, 172-182. http://dx.doi.org/10.1016/j.foodchem.2016.05.143
- Baumes, R. L., Aubert, C. C., Günata, Z. Y., De Moor, W., Bayonove, C. L., and Tapiero, C. (1994). Structures of two C13-norisoprenoid glucosidic precursors of wine flavour. J. Essent. Oil Res. 6, 587–599. doi: 10.1080/10412905.1994.9699350.
- Skouroumounis, G. K., Massy-Westropp, R. A., Sefton, M. A., and Williams, P.J. (1992). Precursors of damascenone in fruit juices. Tetrahedron Lett. 33, 3533–3536. doi: 10.1016/S0040-4039(00)92682
Line 248: Change "tablica"
Answer:
Line 259: We changed "tablica" to “Table”
Line 290: It is quite surprising that the chamber-dried samples are located very far one from the other, why is the variability so large? Quite similar happens with Greenhouse dried grapes. In fact, some of the repetitions are quite close to grapes before drying
Answer:
We agree with the comment that the samples “should be closer” and those differences are due to the great variability of the fresh grapes. However, before the analysis itself, we tested the significance of the difference of all measured parameters with respect to the grape samples. Less than 10% (more precisely 9.8%) of the observed parameters showed a significant difference (p <0.05), however as the tolerated coefficient of variability is 10%, we observed all samples. It is the distribution of the samples within the same quadrant that indicates that the decision is appropriate, but consequently the samples are quite distant from each other.
Conclusion: There is no clear conclusion, only a resume of the results. The differences in the concentration of damascenone and some comments of the practical implications of the findings would clearly improve this section.
Answer:
Thank you for the suggestions. We added in:
Line 351-357: “These findings will be useful in dessert wine Prošek production. Based on our study, greenhouse-dried grapes at the controlled conditions (max. temperature at 37 °C) may provide improved must aroma composition for the production of Prošek wine. In opposite to that, our research showed that temperature of chamber-drying at 50 °C negatively influenced on the concentration of some alcohols, esters and especially on β-damascenone. Greenhouse- dried grapes will contribute to the aroma profile of sweet and fruity descriptors that dominating in scents of over-ripe plums and stewed apple.”

Reviewer 2 Report
Dear Authors,
This paper represents an interesting study of great importance to the wine industry actors and the wine research field.
Only minor aspects need to be revised. Please pay attention to the measuring units of the parameters used. Some are missing (see line 82). Was the drying procedure a common one used in Croatia? If yes, please specify this. If not, then give supporting references for choosing those process parameters. Otherwise, readers will not understand why this procedure (drying in ”a single layer in a hot-air chamber SP-440 C”) is the best one. Explain why did you choose only one temperature variant. Also, specify the duration of each drying process applied.
Author Response
Response to Reviewer 2 Comments
English language and style are fine/minor spell check required
The manuscript was revised in English language by native professional speaker.
Only minor aspects need to be revised. Please pay attention to the measuring units of the parameters used. Some are missing (see line 82). Was the drying procedure a common one used in Croatia? If yes, please specify this. If not, then give supporting references for choosing those process parameters. Otherwise, readers will not understand why this procedure (drying in ”a single layer in a hot-air chamber SP-440 C”) is the best one. Explain why did you choose only one temperature variant. Also, specify the duration of each drying process applied.
Question 1 Please pay attention to the measuring units of the parameters used. Some are missing (see line 82)
Answer:
Thank you. Missing parameter “% “ was added in the line 82.
Question 2: Was the drying procedure a common one used in Croatia?
Line 77-81: We added: “The original technology for Prošek wine making involves drying the grapes, however different drying methods are currently used in Croatia (sun-drying, chamber-drying, greenhouse-drying, etc.). To compare volatile compounds formed in dried grapes under two controlled conditions we set up drying experiment in the greenhouse that is different technique from the direct sun-drying method and the chamber-drying one.”
Question 3: Explain why did you choose only one temperature variant?
Thank you. Line 85-88: We added: “Based on the study Serratosa et al. [19], chamber-drying at 50 °C had advantages over the sun-drying and chamber-drying at 40 °C in terms of aroma, flavor and colour of the must from Pedro Ximenez grapes and therefore it was selected for the investigated temperature of the chamber-drying experiment in the study.”
We added reference 19:
- Serratosa, M.P.; Marquez, A.; Lopez-Toledano, A.; Merida, J. Sensory Analysis of sweet musts in Pedro Ximénez cv. grapes dried using different methods. S. Afr. J. Enol. 2012, 33, 14-20. https://doi.org/10.21548/33-1-1301.
Response to Reviewer 2 Comments
English language and style are fine/minor spell check required
The manuscript was revised in English language by native professional speaker.
Only minor aspects need to be revised. Please pay attention to the measuring units of the parameters used. Some are missing (see line 82). Was the drying procedure a common one used in Croatia? If yes, please specify this. If not, then give supporting references for choosing those process parameters. Otherwise, readers will not understand why this procedure (drying in ”a single layer in a hot-air chamber SP-440 C”) is the best one. Explain why did you choose only one temperature variant. Also, specify the duration of each drying process applied.
Question 1 Please pay attention to the measuring units of the parameters used. Some are missing (see line 82)
Answer:
Thank you. Missing parameter “% “ was added in the line 82.
Question 2: Was the drying procedure a common one used in Croatia?
Line 77-81: We added: “The original technology for Prošek wine making involves drying the grapes, however different drying methods are currently used in Croatia (sun-drying, chamber-drying, greenhouse-drying, etc.). To compare volatile compounds formed in dried grapes under two controlled conditions we set up drying experiment in the greenhouse that is different technique from the direct sun-drying method and the chamber-drying one.”
Question 3: Explain why did you choose only one temperature variant?
Thank you. Line 85-88: We added: “Based on the study Serratosa et al. [19], chamber-drying at 50 °C had advantages over the sun-drying and chamber-drying at 40 °C in terms of aroma, flavor and colour of the must from Pedro Ximenez grapes and therefore it was selected for the investigated temperature of the chamber-drying experiment in the study.”
We added reference 19:
- Serratosa, M.P.; Marquez, A.; Lopez-Toledano, A.; Merida, J. Sensory Analysis of sweet musts in Pedro Ximénez cv. grapes dried using different methods. S. Afr. J. Enol. 2012, 33, 14-20. https://doi.org/10.21548/33-1-1301.
